# Disentangling the mechanistic role of loop-C capping in Cys-loop receptor activation

Gisela D. Cymes [1] & Claudio Grosman [1,2,3] ✉

A comparison of atomic models of Cys-loop receptors reveals that the largest rearrangement upon agonist binding and gating is the contraction ("capping") of the neurotransmitter binding-site loop C. The capping of this loop has often been suggested to act as the mechanical link that couples the binding of extracellular ligands to the gating of the transmembrane pore. However, because binding and gating are inextricably linked, testing this idea experimentally has proved challenging. Here, we disentangle binding and gating using mutagenesis, chimeric constructs, and functional assays. Our results point to the notion that loop-C capping is not required for the pore of Cys-loop receptors to open/desensitize in response to agonist binding to the extracellular domain. Instead, the functional impact of this marked rearrangement seems to be confined to local changes at the level of the neurotransmitter-binding sites, such as the transition from the unliganded state to the low-affinity agonist-bound conformation and/or the transition from the latter to the high-affinity bound state.

Cys-loop receptors are pentameric ligand-gated ion channels consisting of an extracellular domain (ECD) that harbors the neurotransmitter-binding (orthosteric) sites and a transmembrane domain (TMD) that forms the ion-permeable pore (Fig. 1). The ECD orthosteric sites can be unliganded (apo) or bound, and ligands can be bound with, at least, two different affinities ("low" and "high") afforded by different, interconvertible conformations. The TMD can also adopt different conformations, and these either allow (open) or prevent (closed and desensitized) the flow of ions through the membrane. Crucially, five covalent linkers (the pre-M1 linkers; one per subunit) bring the two domains together in such a way that their conformations—and hence, their functional properties—depend tightly on each other's[1]. Indeed, the TMD-pore switches from closed to open/desensitized as the affinity of Cys-loop receptors for orthosteric agonists switches from low to high. Therefore, the term "binding–gating coupling" used throughout this paper should not be taken literally. Certainly, it is not the mere act of ligand binding that is coupled to the gating of the pore, but rather, it is the low-affinity ⇌ high-affinity interconversion of the agonist-bound orthosteric sites that is tied to the closed ⇌ open/desensitized rearrangement of the TMD[1,2]. It is

precisely this "linkage" or "coupling" of ligand-binding and effector domains that underlies the physiological role of all ionotropic receptors as transducers of extracellular chemical signals into the cell.

A comparison of atomic models of Cys-loop receptors with and without bound agonists reveals that, at the level of the extracellular domain (and with only a few exceptions[3,4]), the largest rearrangement is the "capping" or "closure" of all (in homomeric channels; e.g.[5–9]) or a subset (in heteromeric channels; e.g.[10–13]) of the loops C. Loop C is the β-hairpin loop between ECD strands 9 and 10 of each subunit (Fig. 1c), and "capping" refers to its displacement toward the long axis of the receptor-channel in going from the unliganded closed state to the agonist-bound open or desensitized state. Comparing these two end states, the tip of loop C moves >10 Å (measured at the Cα atoms upon global superposition) in the homomeric α7 nicotinic acetylcholine receptor (AChR[7,9]); ~9–10 Å in the muscle-type AChR[11–13]; ~7 Å in the heteromeric α1β3γ2L γ-aminobutyric-acid-type-A receptor (GABA_AR[10]); and ~5 Å in the homomeric α1 glycine receptor (GlyR[6]), for example; Fig. 2 illustrates this phenomenon with atomic models of the α7 AChR.

[1]Department of Molecular and Integrative Physiology, University of Illinois Urbana-Champaign, Urbana, IL, USA. [2]Center for Biophysics and Quantitative Biology, University of Illinois Urbana-Champaign, Urbana, IL, USA. [3]Neuroscience Program, University of Illinois Urbana-Champaign, Urbana, IL, USA. ✉e-mail: grosman@illinois.edu

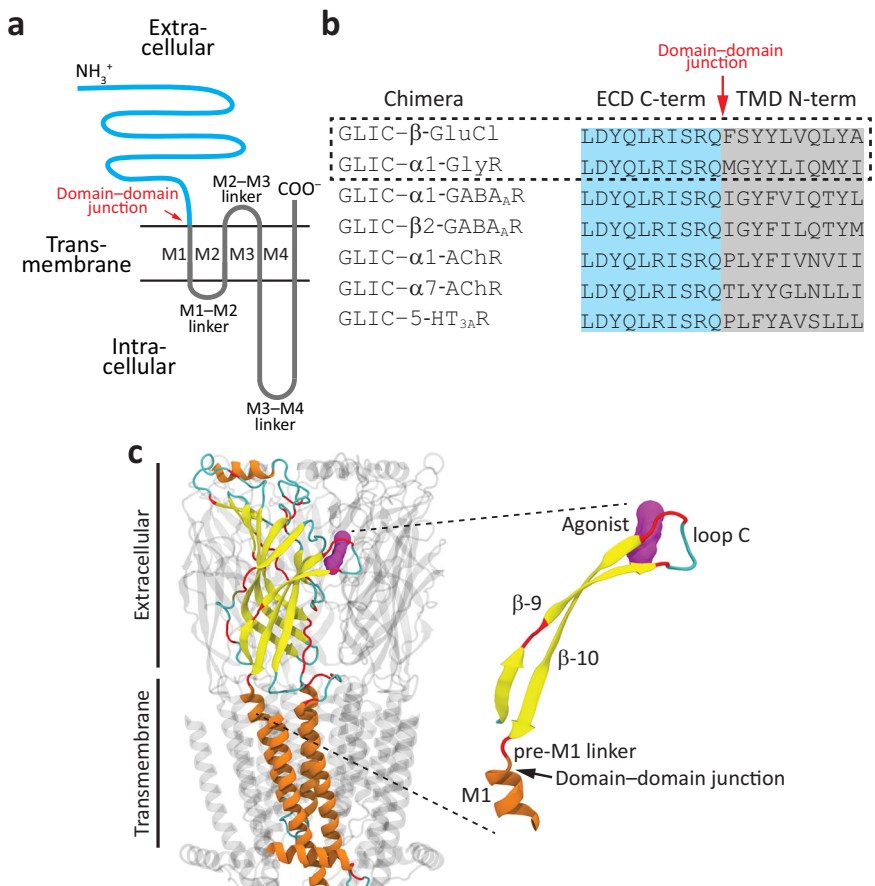

**Fig. 1 | Activating the pore of animal Cys-loop receptors with protons.**
**a** Membrane-threading pattern common to all Cys-loop-receptor subunits. The ECD is shown in cyan, and the TMD, in gray. **b** Amino-acid sequences at either side of the domain–domain junction of the seven ECD–TMD chimeras tested here. The two chimeras that gave rise to the largest macroscopic currents—and thus, those for which the effect of deleting all five loops C on function was studied—are indicated with a dashed-line rectangle. **c** Atomic model of a Cys-loop receptor (agonist-bound desensitized human α7 AChR; PDB ID 7KOQ[7]). One of the subunits is color-coded on the basis of secondary structure (α-helix: orange; β-strand: yellow; $3_{10}$-helix: blue; turn: dark cyan; and coil: red)[59], and the molecule of bound orthosteric agonist (epibatidine) is colored purple. The molecular images were made with Visual Molecular Dynamics (VMD)[57] using cartoon representation for the protein and surface representation for the agonist. For the sake of clarity, the C-terminal tail of the highlighted subunit was omitted, and the intracellular portion of the channel is not shown.

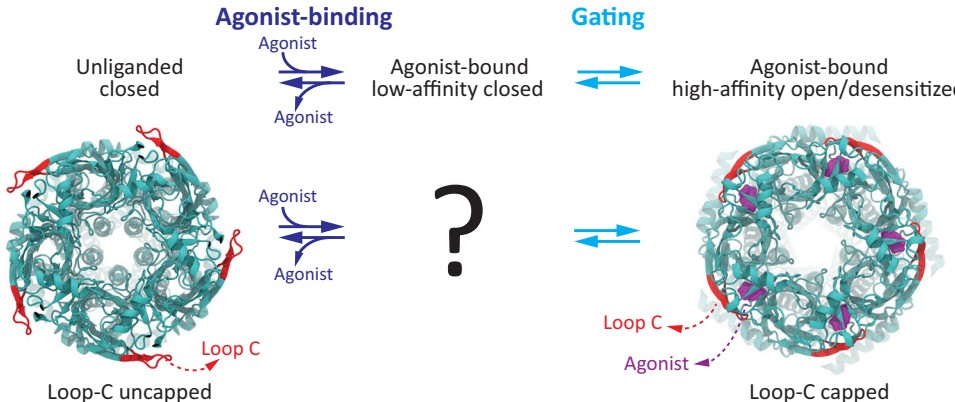

**Fig. 2 | The capping of loop C in homomeric Cys-loop receptors.** Atomic models of the unliganded-closed (PDB ID 7EKI[9]) and agonist-bound open (PDB ID 7KOX[7]) human α7 AChR. The view is perpendicular to the membrane from the extracellular side. Loops C are colored red; bound agonist molecules (epibatidine), purple; and all other atoms, dark cyan. The structure of the agonist-bound low-affinity complex—that is, the well-established conformation that lies between the unliganded-closed receptor and the high-affinity complex—remains unknown. Atomic models proposed to correspond to additional intermediate conformations between the unliganded and agonist-bound open/desensitized states have recently been reported for a variety of Cys-loop receptors embedded in lipid nanodiscs (e.g.[60,61]). However, the functional state(s) these intermediates represent remain uncertain. The capping of loop C illustrated here on an atomic model of the homomeric α7 AChR is representative of that occurring upon the binding of small-molecule agonists and gating in other members of the superfamily. The molecular images were made with VMD[57] using cartoon representation for the protein and surface representation for the agonist.

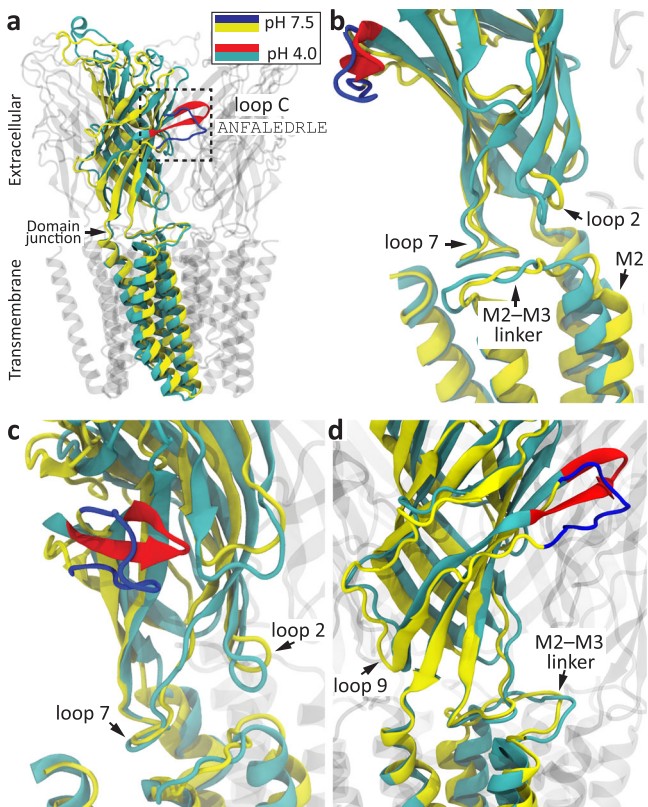

**Fig. 3 | Structural rearrangements upon proton binding and gating in GLIC. a–d** Different views of globally superposed atomic models of closed (pH 7.5; PDB ID: 8I42[34]) and open (pH 4.0; PDB ID: 8WCR[34]) GLIC. One subunit is highlighted, and its loop C (sequence: ANFALEDRLE) is colored blue (pH 7.5) or red (pH 4.0); all other atoms are colored yellow (pH 7.5) or dark cyan (pH 4.0). Different structural elements at the ECD–TMD interface are indicated. The functional role of loop-C capping in GLIC remains enigmatic inasmuch as deleting all five loops C (from full-length GLIC)—much as we did here in the context of ECD–TMD chimeras—had, essentially, no effect on proton-gated currents[28]. Compared to small-molecule gated Cys-loop receptors from animals, GLIC has a much shorter M3–M4 linker. The molecular images were made with VMD[57] using cartoon representation.

Because loop-C residues contribute to the orthosteric ligand-binding sites[14], this capping rearrangement has frequently been interpreted as the conformational change that underlies the low-affinity → high-affinity transition of the ECD. Moreover, because of its location—squarely between the binding sites and the first transmembrane segments (M1)—the capping of this loop has quite sensibly been suggested to act as the mechanical link that couples the low-to-high agonist-affinity change of the ECD to the gating of the TMD pore (e.g.[11,15–21]). The picture that emerges is one where the capping of loop C, in addition to trapping the ligand in the high-affinity bound state, causes the repositioning of the extracellular end of the adjacent M1 α-helix. In turn, the latter would initiate a domino-like cascade of rearrangements at the ECD–TMD interface that, eventually, culminates in pore opening and desensitization. Structural details aside, the implications of this model are clear: without loop-C capping, agonist-driven pore gating would not be possible. Domain–domain communication would be severed, and thus, the coupling between the ECD and the TMD would be lost.

Although the direct contribution of loop-C residues to the binding of orthosteric ligands has been clearly established[14], the suggestion of a role for its capping rearrangement in the low-to-high affinity change of the orthosteric sites or the coupling between the latter and pore gating has not been tested experimentally. In this work, we focus on the involvement of loop-C capping in domain–domain coupling.

Concretely, we ask whether the pore of agonist-bound Cys-loop receptors requires the capping of loop C to gate. Because loop-C capping underlies most of the reorganization of the orthosteric sites (and the ECD as a whole) upon agonist binding and gating, the importance of understanding what this conformational change does—and what it does not do—can hardly be overemphasized.

Undoubtedly, the most compelling way to tackle this question would be to delete the loops C of a Cys-loop receptor and probe for function on application of agonists. However, because these loops (all of them in homomeric channels, some of them in heteromeric channels) are necessary for binding agonists in the first place, agonist-activated currents would not be elicited from such a construct, and therefore, the question could not be answered. It is not surprising, then, that the role of loop-C capping in the binding–gating coupling of Cys-loop receptors has remained difficult to ascertain (e.g.[22–26]). In fact, to our knowledge, the effects of loop-C perturbations on gating—without the confounding effects of altered ligand binding—have thus far only been studied by omitting the ligand altogether. Unfortunately, however, published reports on the requirement of loop-C capping for unliganded gating have been conflicting[22,23,26].

Here, to disentangle the contribution of loop-C residues to the binding of orthosteric ligands, on the one hand, and to the binding–gating-coupling phenomenon, on the other, we set out to work with GLIC[27]. GLIC is a member of the superfamily that gates upon binding ligands (protons) to ECD residues other than those in loop C[28,29] (Supplementary Fig. 1). Importantly, the three-dimensional structure of this bacterial channel is essentially the same as that of its orthologs from animals[30,31], and so are the rearrangements that connect the unliganded closed state to the agonist-bound open/desensitized state[32–34]. Indeed, in GLIC, lowering the pH to channel-activating values leads to the repositioning of structural elements at the ECD–TMD interface (loops 2, 7, and 9 on the ECD side, and the M2–M3 linker on the TMD side) and the outward pulling of the extracellular end of the M2 α-helix (Fig. 3) in much the same way as has been observed for the Cys-loop receptors gated by small molecules; Fig. 4 illustrates the latter rearrangements using a heteromeric GABA$_A$R as an example. Notably, the similarities extend to GLIC's loop C, which undergoes a rearrangement that is strikingly reminiscent of loop-C capping.

We reason that, if the capping of loop C were required for agonist-driven pore gating, then deleting this loop would result in a total loss-of-function phenotype. Because we are not interested in the gating of the pore of GLIC, but rather, in the gating mechanism of its neurotransmitter-gated counterparts, we fuse the ECD of GLIC to the TMD of Cys-loop receptors from animals and test the effect of deleting the entire loop C (of all five subunits) on function. The results presented here suggest that agonist-driven pore gating does not require the capping of loop C. Therefore, loop-C capping is unlikely to play a role in the domain–domain communication that couples conformational changes in the ECD to those occurring in the TMD (and vice versa).

We also address, here, the related question as to the role of loop-C residues of heteromeric-channel-forming subunits whose loops C do not contribute to the orthosteric sites. To this end, we resort to the muscle AChR and find that the application of ACh to channels lacking the loop C of the β1, δ or ε subunit elicits currents of broadly wild-type characteristics.

## Results

### Disentangling ligand binding and binding–gating coupling

cDNAs coding homomeric chimeras consisting of the ECD of GLIC and the TMD of several Cys-loop-receptor subunits from animals were transiently transfected in HEK-293 cells, and the currents elicited by extracellular acidification were recorded. The different TMDs corresponded to those of human α7 AChR, α1 GABA$_A$R, β2 GABA$_A$R, and α1 GlyR; mouse α1 AChR and 5-hydroxytryptamine type 3A receptor (5-

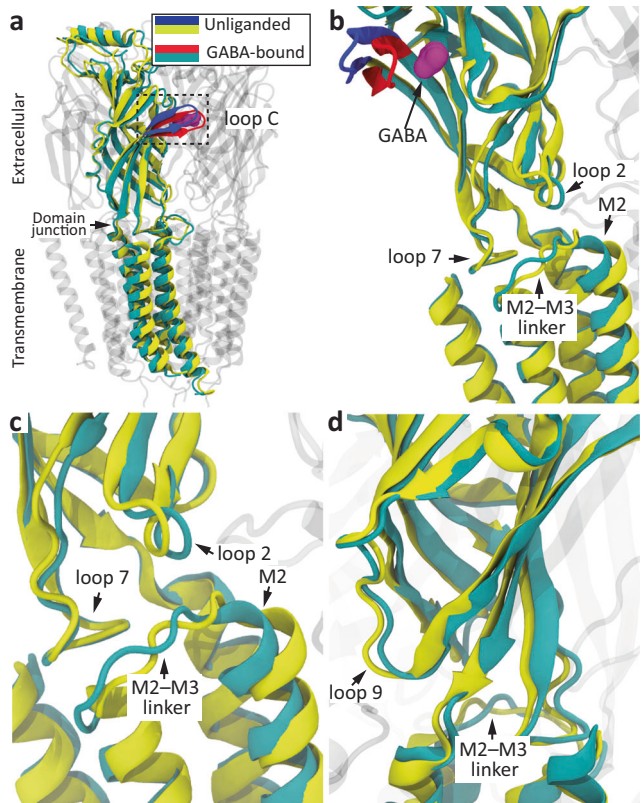

**Fig. 4 | Structural rearrangements upon GABA binding and gating in a heteromeric GABA_AR. a–d** Different views of globally superposed atomic models of closed (unliganded orthosteric sites; PDB ID: 6HUG[10]) and desensitized (orthosteric sites bound to GABA; PDB ID: 6HUO[10]) $(\alpha 1)_2(\beta 3)_2\gamma 2L$ GABA_AR. One subunit is highlighted, and its loop C is colored blue (unliganded) or red (GABA-bound); bound GABA is colored purple; and all other atoms are colored yellow (unliganded) or dark cyan (GABA-bound). In (**a–c**), the highlighted subunit is one of the β3 subunits. In (**d**), the highlighted subunit is one of the α1 subunits. Different structural elements at the ECD–TMD interface are indicated. The structural rearrangements of loop C and elements of the ECD–TMD interface illustrated here on an atomic model of a heteromeric GABA_AR are representative of structural rearrangements occurring upon the binding of small-molecule agonists and gating in other members of the superfamily. The molecular images were made with VMD[57] using cartoon representation for the protein and surface representation for the agonist.

HT_3AR); and *C. elegans* β glutamate-gated Cl⁻ channel (β GluCl). All seven chimeric subunits were simple "cut-and-splice" constructs with no additional mutations (Fig. 1a, b). Fast-perfusion patch-clamp experiments revealed that the GLIC–β-GluCl chimera generated the largest currents, followed by the GLIC–α1-GlyR construct. Thus, we focused on these two best-expressing chimeras for the study of the effects of loop-C deletion on pore gating.

Figure 5a, b shows whole-cell inward currents recorded from the "wild-type" GLIC–β-GluCl chimera and its loop-C-deleted counterpart; the currents were elicited by the extracellular application of low-pH pulses (7.4 → 4.5 → 7.4) of different durations. To minimize the contribution of proton-activated currents mediated by ASIC-1a endogenously expressed in HEK-293 cells[35], all extracellular solutions contained 200-μM amiloride; amiloride's half-inhibition concentration ($IC_{50}$) of endogenous ASIC-mediated currents was reported to be 2.2 μM[35]. Also, to minimize the contribution of endogenous currents mediated by the proton-activated, outwardly rectifying anion channel[36], 4,4′-diisothio-cyanostilbene-2,2′-disulphonic (DIDS) was also added to the extracellular solutions (final concentration: 100 μM; $IC_{50} \sim 2.9$ μM[36]) whenever outward currents were recorded (Fig. 5c). Notably, we found that the deletion of all five loops C from GLIC (sequence: ANFALEDRLE)

did not prevent the gating of the chimera's β-GluCl pore in response to low-pH pulses. Actually, the kinetics of activation, deactivation (Fig. 5a) and desensitization (Fig. 5b) of the deletion mutant were not very different from those displayed by the wild-type construct, which is remarkable considering that the mutant lacks 10 residues per subunit (for a total of 50 residues) only 12 residues away from the ECD–TMD covalent junction (Fig. 3a). Also, these kinetics are comparable to those displayed by naturally occurring members of the superfamily under the same experimental conditions (see Fig. 5c for responses to brief agonist pulses), a finding that lends further credence to the use of the chimeric approach reported here. In addition, GLIC–β-GluCl retained the nearly perfect anion-selectivity of full-length β GluCl[37], even when the ECD of the chimera corresponded to that of a cation-selective member of the superfamily (Fig. 5d). This observation is well-aligned with the notion that, to the extent that it can be measured, the properties of the ECD have no effect on the charge selectivity of Cys-loop receptors (e.g.[38,39]).

To extend our observations to other members of the superfamily, we also recorded currents from the GLIC–α1-GlyR chimera. As was the case for the construct containing the *C. elegans* β-GluCl TMD, deleting all five loops C did not prevent the opening or desensitization of the human α1-GlyR pore in response to protons (Fig. 6).

**Allosteric loops C of heteromeric receptors—the other loops C**
In heteromeric members of the superfamily, only a subset of loops C (referred here to as "orthosteric loops C") contributes to the formation of the orthosteric neurotransmitter-binding sites and cannot be deleted without compromising agonist binding. The other loops C ("allosteric loops C"), however, are not needed for the binding of agonists, and thus, in principle, their role in ECD–TMD coupling can be probed with mutagenesis without the need of resorting to chimeric constructs. Although the extent to which the allosteric loops C rearrange upon agonist binding and gating seems to be very small[10–13] (Fig. 7), a role in domain–domain coupling could be expected from every single loop C because the low-to-high affinity change of the ECD needs to be transmitted to the transmembrane portion of all five subunits for the pore to open and desensitize. Moreover, studies of neurotoxins that bind to the allosteric loops C of neuronal AChRs[40] or GABA_ARs[41] revealed that toxin binding has clear effects on agonist-evoked currents, further hinting at the possibility of a role for these loops in the communication between the ECD and the TMD. To address this point, we set out to delete the loops C of the β1 (GDQRGGKEGHHE), δ (PSVPMDSTNHQD), and ε (YEGGSTEGPGET) subunits of the adult-type muscle AChR; those of the two α1 subunits (that is, the orthosteric loops C) were left unaltered.

Deleting all three allosteric loops C prevented the expression of the receptor on the plasma membrane, as revealed by the absence of specific binding of membrane-impermeant [125I]-α-bungarotoxin to the cell surface of transfected HEK-293 cells. Therefore, we proceeded to express muscle-AChRs containing a single loop-C-deleted subunit at a time (β1, δ or ε, the other four being wild-type). A concern that often arises when engineering mutations in single subunits of heteromeric Cys-loop receptors is the possibility that the assembled pentamers lack the mutant subunit altogether. Certainly, although the stoichiometry of the adult-muscle AChR is two α1, and one each of β1, δ, and ε subunits, any non-α subunit can be replaced by another subunit, particularly if (as was the case here) the biogenesis of the mutated subunit is adversely affected. Furthermore, muscle AChRs of aberrant subunit compositions give rise to currents with unremarkable kinetics in response to the application of agonist (Fig. 8; see also refs. 42,43) and thus, their expression on the plasma membrane could be highly misleading. Hence, to make sure that the loop-C-deleted subunits were incorporated into the pentamers, we engineered a second mutation in the background of the deletion: a Leu-to-His mutation in the middle of the pore-lining α-helix M2 (Fig. 1a), at position 9′. As we have previously

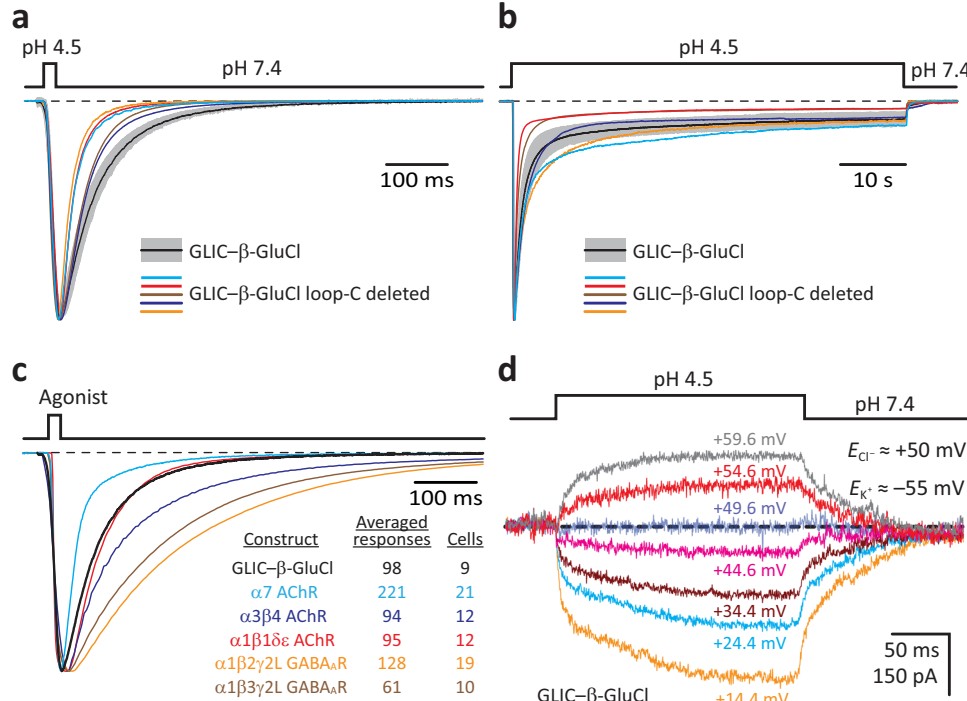

**Fig. 5 | Agonist-driven gating of β-GluCl's pore does not require the capping of loop C. a**, **b** Inward currents recorded from the indicated constructs in the whole-cell patch-clamp configuration in response to 20-ms (**a**) or 1-min (**b**) exposures to pH-4.5 extracellular solution. For the "wild-type" GLIC–β-GluCl ECD–TMD chimera, responses are shown as the mean (black solid line) ± one standard deviation (SD; gray error bars) of normalized responses recorded from different whole-cell experiments. Averaged wild-type data in (**a**) correspond to a total of 98 responses recorded from 9 different cells, and those in (**b**) correspond to a total of 9 responses recorded from 9 different cells. Each individual normalized response shown for the loop-C-deleted mutant was recorded from a different cell. The pipette potential was −60 mV. **c** Inward currents recorded from the indicated constructs in the whole-cell patch-clamp configuration in response to 20-ms pulses of agonist. The latter was pH 4.5 for the GLIC–β-GluCl chimera; 100-μM ACh for both the α7 AChR and the α1β1δε AChR; 200-μM ACh for the α3β4 AChR; and 1-mM

GABA for both heteromeric GABA$_A$Rs. Each displayed trace is the average of several normalized responses recorded from different cells; for the sake of clarity, only the mean of each distribution is shown. The traces were horizontally aligned at their half-activation times. The chimera's currents decay (deactivate) more slowly than those of the α7 AChR, faster than those of the ganglionic (α3β4) AChR and heteromeric GABA$_A$Rs, and at nearly the same rate as those of the adult-type muscle (α1β1δε) AChR. The pipette potential was −60 mV. **d** Whole-cell currents recorded under KCl-dilution conditions (high KCl concentration in the pipette solution; low, in the extracellular solution); the pipette potentials indicated next to each trace were corrected for the liquid-junction potential that arises between these two solutions. The indicated equilibrium potentials ($E_{Cl^-}$ and $E_{K^+}$) were calculated at 22 °C using ion concentrations. The reversal potential of the chimera was very close to the equilibrium potential for Cl$^-$ ($E_{Cl^-}$). For all panels, black dashed lines denote the zero-current baseline. Source data are provided as a Source Data file.

shown using single-channel recordings from the muscle AChR, the alternate protonation and deprotonation of single pore-lining ionizable side chains manifests as pH-sensitive fluctuations of the cation currents between two open-channel levels[44–46]. Here, we chose to replace the leucines at position 9′ with histidines because the bulk-like p$K_a$ of the pore-facing imidazole side chain and the partial extent of current block that follows its protonation allowed the unambiguous detection of the loop-C-deleted subunit in the pentamer even at pH 7.4. In addition, the Leu-to-His mutation at position 9′ slows down both channel closing and entry into desensitization[44], which further facilitates the identification of the presence of the loop-C deletion.

Figures 7 and 9 show macroscopic and single-channel current traces, respectively, recorded from the wild-type muscle AChR and its (single-subunit) loop-C-deleted counterparts. The single-channel data clearly show that the mutant subunits are incorporated into the channel and that allosteric-loop-C deletions in single subunits are functionally well-tolerated. Furthermore, the slower kinetics of macroscopic-current decay recorded from the mutants (Fig. 7b, c) indicate that the loop-C deleted + L9′H subunits express well and are present in a sizable fraction—perhaps, all—of the plasma-membrane AChRs. Evidently, we cannot rule out the possibility that some subunit-omitted receptors expressed and contributed to the macroscopic currents. However, since currents through subunit-omitted muscle AChRs do not decay more slowly (Fig. 8), these aberrant AChRs could

at most account for only a small fraction of the expressed AChRs. It follows, then, that any single allosteric loop C is largely dispensable for the expression, assembly, and function of the muscle AChR.

## Discussion

Addressing the question as to whether the capping of the orthosteric-site loop C couples the low ⇌ high agonist-affinity change of the ECD to the closed ⇌ open/desensitized rearrangement of the TMD is challenging because these loops are part of the orthosteric agonist-binding sites, and thus, their mutation ends up interfering with the very phenomenon we seek to understand. It was with this critical limitation in mind that we resorted to GLIC, a member of the superfamily that is efficaciously gated by protons binding to ionizable side chains scattered throughout the ECD[28,29] (Supplementary Fig. 1), and therefore, whose five loops C can be deleted without abolishing the binding of agonist. Although this bacterial channel may have lost some of its appeal as a model system for the structural biology of Cys-loop receptors from animals, the fact that GLIC retains all the functional properties of a genuine member of the superfamily[27] without using a canonical orthosteric ligand-binding site renders its ECD a most powerful tool when it comes to dissecting activation mechanisms. Indeed, for binding-site residues, participation in binding and gating are inextricably entwined, and the use of GLIC's ECD in chimeric constructs allows a clear separation of these two phenomena.

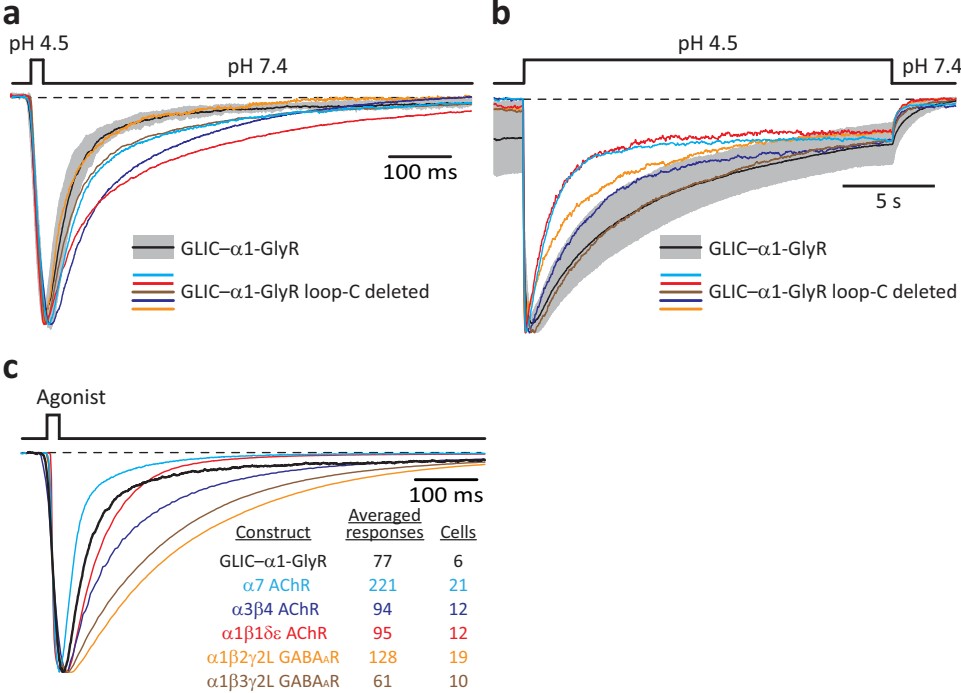

**Fig. 6 | Agonist-driven gating of the α1-GlyR's pore does not require the capping of loop C. a, b** Inward currents recorded from the indicated constructs in the whole-cell patch-clamp configuration in response to 20-ms (**a**) or 20-s (**b**) exposures to pH-4.5 extracellular solution. For the "wild-type" GLIC–α1-GlyR ECD–TMD chimera, responses are shown as the mean (black solid line) ± 1 SD (gray error bars) of normalized responses recorded from different whole-cell experiments. Averaged wild-type data in (**a**) correspond to a total of 77 responses recorded from 6 different cells, and those in (**b**) correspond to a total of 11 individual responses recorded from 11 different cells. Each individual normalized response shown for the loop-C-deleted mutant was recorded from a different cell. The pipette potential was −60 mV. In (**a**), the black dashed line denotes the current level at pH 7.4 at equilibrium, which (especially for the wild-type construct) was not zero. In (**b**), the black dashed line denotes the zero-current baseline. **c** Inward currents recorded from the indicated constructs in the whole-cell patch-clamp configuration in response to 20-ms pulses of agonist. The latter was pH 4.5 for the GLIC–α1-GlyR chimera; 100-μM ACh for both the α7 AChR and the α1β1δε AChR; 200-μM ACh for the α3β4 AChR; and 1-mM GABA for both heteromeric GABA$_A$Rs. Each displayed trace is the average of several normalized responses recorded from different cells; for the sake of clarity, only the mean of each distribution is shown. The traces were horizontally aligned at their half-activation times. This chimera's deactivation kinetics are similar to those of the GLIC–β-GluCl construct (Fig. 5) and the adult-type muscle AChR. The pipette potential was −60 mV. The black dashed line denotes the zero-current baseline. Source data are provided as a Source Data file.

Our results point to the notion that the capping rearrangement of loop C is not required for the pore of small-molecule-gated Cys-loop receptors to open/desensitize in response to agonist binding. It follows, then, that the functional impact of loop-C capping must be confined to local changes at the level of the orthosteric sites such as the transition of the ECD from the unliganded state to the low-affinity-bound conformation and/or the transition from the latter to the high-affinity bound state (Fig. 2). It further follows that it is the more subtle repositioning of some other structural element of the orthosteric sites that couples the low-to-high affinity transition of the agonist–receptor complex to the opening/desensitization of the closed pore. We hypothesize that these conclusions extend to all small-molecule-gated members of the superfamily. Indeed, it seems unlikely that such a fundamental aspect of ligand-gated ion-channel operation could hold true only for the pores of β-GluCl and the α1-GlyR that we studied here. After all, atomic models of Cys-loop receptors in closed, open, and desensitized states have indicated that the structural aspects of ECD–TMD coupling are well-conserved throughout the superfamily[5–13]. Remarkably—and directly relevant to our reasoning here—these commonalities persist in GLIC, inasmuch as the same type of rearrangements occur at the domain–domain interface of these distantly related proton-gated channels upon agonist binding and gating (Figs. 3 and 4). In other words, regardless of whether it is protons binding to a fuzzy array of ionizable side chains or small-molecule agonists associating with a few well-defined binding pockets, domain–domain communication seems to proceed through a largely conserved structural mechanism (see also ref. 47 for a similar conclusion). Thus, although

the (necessarily) indirect nature of our experimental approach does not allow us to rule it out completely, the possibility that our conclusions pertain only to proton-gated constructs of the superfamily seems also very unlikely. Moreover, note that our findings do not contradict at all the experimental observation that mutations to loop C affect the kinetics of agonist-evoked currents through Cys-loop receptors (e.g.[19,21,48]). Indeed, because agonist-driven gating depends directly on agonist affinities[49–54], changes in the rate and equilibrium constants of liganded gating would also be expected from mutations to residues that only contribute to ligand binding. Binding and gating are mechanistically linked.

Our current understanding of the structure of the low-affinity agonist-bound conformation of Cys-loop receptors (that is, the intermediate state in Fig. 2) is so limited that the extent to which loop-C capping takes place upon agonist binding with low affinity to the closed state *versus* as a result of the low-to-high affinity conformational change that is somehow coupled to the opening/desensitization of the transmembrane pore remains unclear. It is worth noting here that the practice of structurally aligning the unliganded (apo) and agonist-bound atomic models of Cys-loop receptors most often compares closed and open/desensitized states, respectively, thus skipping altogether the crucial conformation that lies in between. A practical consequence of this incomplete understanding is that we cannot yet infer the affinity with which an agonist is bound to the orthosteric sites from the observation of structures alone. In other words, just because the loops C of an atomic model look capped does not necessarily mean that the ligand–receptor complex is in its high-affinity form; insight

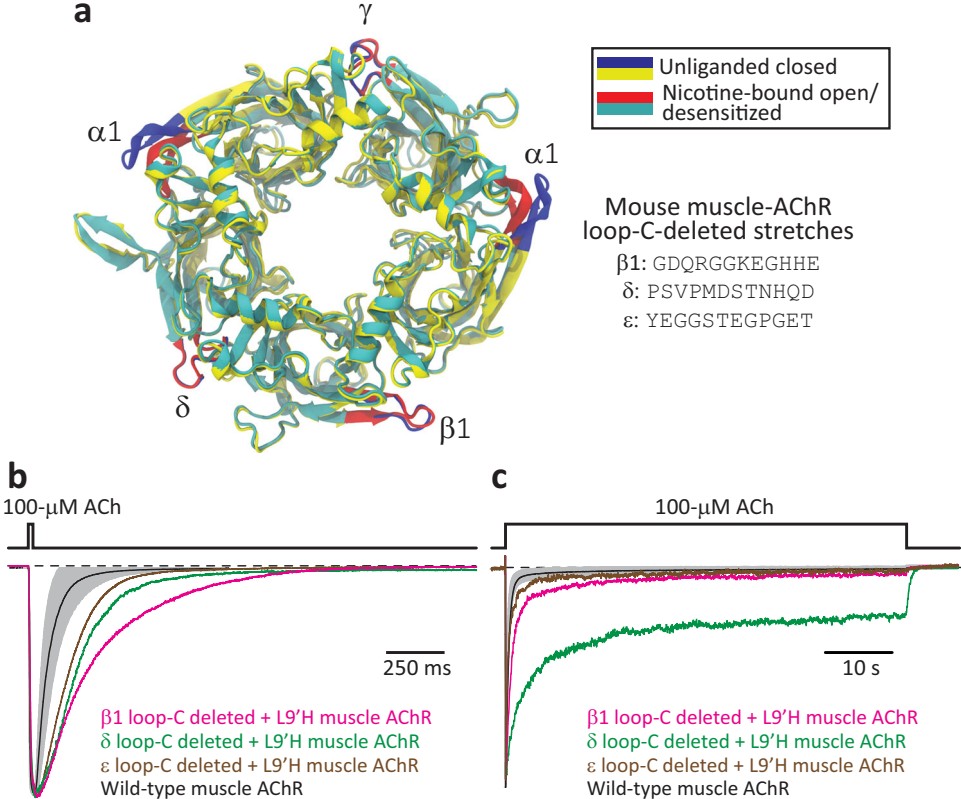

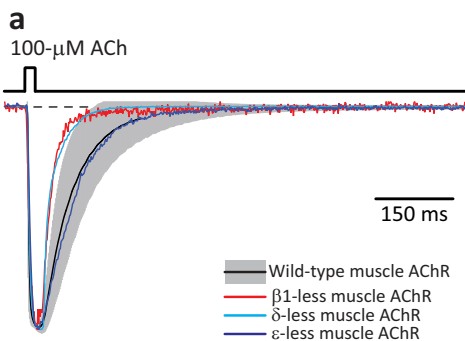

**Fig. 7 | Macroscopic currents from the muscle AChR lacking allosteric loops C.**
**a** Global superposition of atomic models of closed (unliganded; PDB ID: 7QKO[11]) and open/desensitized (nicotine-bound; PDB ID: 7QL5[11]) muscle-type AChR from *Torpedo californica*. The view is perpendicular to the membrane from the extracellular side. Loop-C atoms are colored blue (unliganded) or red (nicotine-bound); all other atoms are colored yellow (unliganded) or dark cyan (nicotine-bound). For the sake of clarity, neither the TMDs nor the molecules of bound nicotine are shown. Upon the binding of nicotine, the loops C of the α1 subunits rearrange much more extensively than those of the non-α-subunits. The sequences of the loops C of the adult mouse-muscle AChR's β1, δ, and ε subunits are indicated (note the presence of an ε subunit in place of the γ subunit of *Torpedo*'s AChR). The molecular images were made with VMD[57] using cartoon representation. **b, c** Inward currents recorded from the indicated constructs in the whole-cell patch-clamp configuration in response to 20-ms (**a**) or 1-min (**b**) exposures to 100-µM ACh. For the wild-type mouse-muscle AChR (adult type), responses are shown as the mean (black solid line) ± 1 SD (gray error bars) of normalized responses recorded from different whole-cell experiments. Averaged wild-type data in (**b**) correspond to a total of 95 responses recorded from 12 different cells, and those in (**c**) correspond to a total of 10 individual responses recorded from 10 different cells. Each normalized trace shown for the loop-C-deleted mutants corresponds to a single response. In addition to introducing a proton-binding site, the reporter mutation (Leu-to-His at position 9′ of the M2 α-helix of the same subunit bearing the deletion of loop C) slows down the timecourses of deactivation (**b**) and entry into desensitization (**c**). The pipette potential was −60 mV. Black dashed lines denote the zero-current baseline. Source data are provided as a Source Data file.

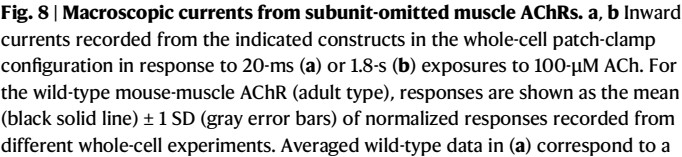

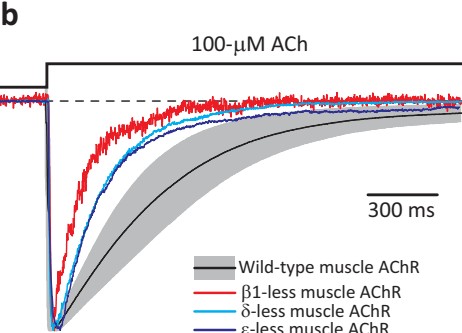

**Fig. 8 | Macroscopic currents from subunit-omitted muscle AChRs. a, b** Inward currents recorded from the indicated constructs in the whole-cell patch-clamp configuration in response to 20-ms (**a**) or 1.8-s (**b**) exposures to 100-µM ACh. For the wild-type mouse-muscle AChR (adult type), responses are shown as the mean (black solid line) ± 1 SD (gray error bars) of normalized responses recorded from different whole-cell experiments. Averaged wild-type data in (**a**) correspond to a total of 95 responses recorded from 12 different cells, and those in (**b**) correspond to a total of 10 individual responses recorded from 10 different cells. Each normalized trace shown for the subunit-omitted mutants corresponds to a single response. The pipette potential was −60 mV. Black dashed lines denote the zero-current baseline. Source data are provided as a Source Data file.

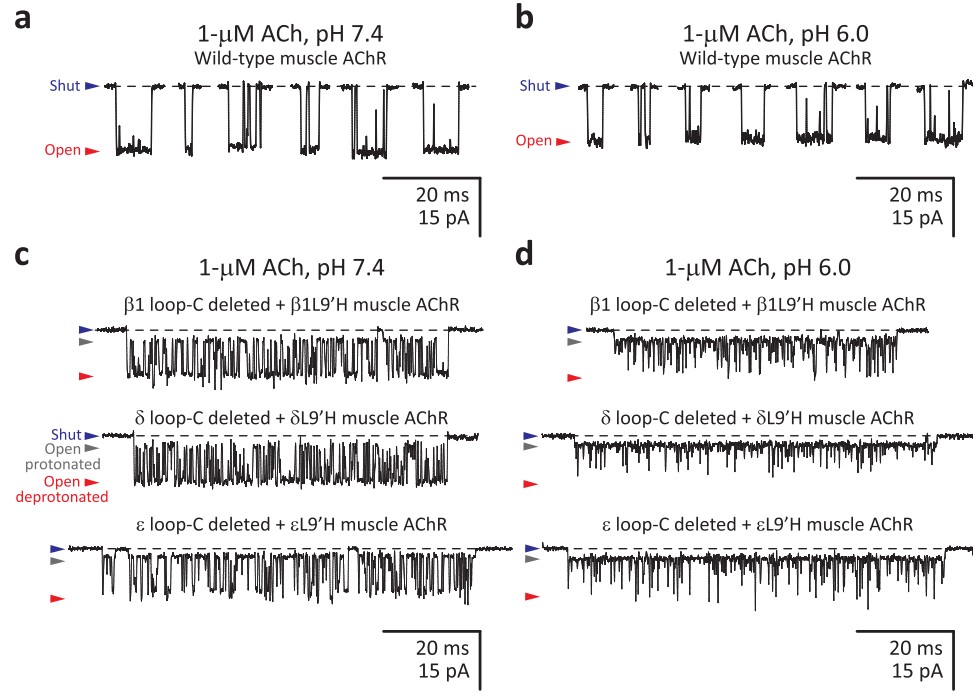

**Fig. 9 | Single-channel currents from the muscle AChR lacking allosteric loops C. a, b** Bursts of single-channel openings recorded in the cell-attached configuration from the wild-type adult mouse-muscle AChR; as previously reported, the single-channel conductance decreased somewhat as the pH was lowered[44]. **c, d** Bursts of single-channel openings recorded in the cell-attached configuration from mutants carrying the indicated mutations in the β1, δ or ε subunit. The reporter mutation (Leu-to-His at position 9′ of the M2 α-helix) introduced a pore-lining proton-binding side chain whose protonation lowers the single-channel conductance, and therefore, manifests as a current sublevel whose occupancy depends on pH[44-46]. Because of the lumen-facing orientation of the engineered histidines in the open-channel conformation of the (cation-selective) muscleAChR, their side-chain $pK_a$s are bulk-like[44], and thus, side-chain

protonation–deprotonation events could be clearly observed, even at pH 7.4. The occurrence of these main-level ⇌ sublevel current fluctuations allowed us to detect the presence of the loop-C-deleted subunit in the pentameric assembly. As expected[62] from the slower deactivation timecourses recorded upon application of a high concentration of ACh (100 µM; Fig. 7b), bursts of single-channel openings recorded from the mutants at a low concentration of ACh (1 µM) were on average longer than those recorded from the wild-type muscle AChR. For all panels, the indicated pH values correspond to those of the pipette solution; openings are downward deflections; the pipette potential was –100 mV; and—for display purposes—the traces were low-pass filtered at 5 kHz. Black dashed lines denote the zero-current baseline. Source data are provided as a Source Data file.

from some type of ligand-binding assay is still needed. Evidently, more work will be required to fully understand the structural bases of the alternative affinities of Cys-loop receptors for agonists. We envision that a thorough understanding of this central aspect of receptor-channel operation will prove crucial for both the rational design of drugs and the correct assignment of functional states to solved structures.

Regarding the allosteric loops C of heteromeric receptors, although previous reports[40,41] suggested to us that their rearrangement upon agonist binding and gating—however small—may be indispensable for function, our experimental observations with the muscle AChR clearly indicate otherwise.

## Methods
### cDNA clones, heterologous expression, and structural comparisons
Complementary DNAs (cDNAs) coding ECD–TMD chimeras were synthesized and subcloned in pcDNA3.1 (GenScript, Invitrogen); the amino-acid sequences at the respective ECD–TMD junctions are indicated in Fig. 1b. cDNA coding the human α7 AChR (UniProt accession number: P36544) in pcDNA3.1 was purchased from addgene (#62276). cDNA coding isoform 1 of human RIC-3[55] (accession number: Q7Z5B4) in pcDNA3.1 was provided by W. N. Green (University of Chicago, IL). cDNA coding human NACHO[56] (TMEM35A; accession number: Q53FP2) in pCMV6-XL5 was purchased from OriGene Technologies Inc. (#SC112910). cDNAs coding the mouse α1, β1, δ, and ε subunits of the (muscle) AChR (accession numbers: P04756, P09690, P02716, and

P20782, respectively) in pRBG4 were provided by S. M. Sine (Mayo Clinic, Rochester, MN). cDNAs coding the human α3 and β4 subunits of the (ganglionic) AChR (accession numbers: P32297 and P30926, respectively) were purchased from horizon and were subcloned in pcDNA3.1. And cDNAs coding the human α1, β2, β3, and γ2L subunits of heteromeric GABA$_A$Rs (accession numbers: P14867, P47870, P28472, and P18507, respectively) in pcDNA3.1 were provided by C. C. Hernandez (Northwestern University) and R. L. Macdonald (Vanderbilt University Medical Center). The different constructs were heterologously expressed in transiently transfected adherent HEK-293 cells (ATCC; CRL-1573) grown at 37°C and 5% $CO_2$ in 35-mm cell-culture dishes. Transfections (performed using a calcium-phosphate-precipitation method) proceeded for 16–18 h. For the expression of the chimeric constructs (wild-type or mutants), we used 1.5 (GLIC–β-GluCl) or 3.0 (GLIC–α1-GlyR) µg cDNA/dish. For the expression of the mouse-muscle AChR (wild-type or mutants), we used a total of 0.75 µg/dish of cDNAs coding the α1, β1, δ, and ε subunits in a 2:1:1:1 ratio (by weight). For the expression of "subunit-less" muscle AChRs, the cDNA coding the omitted subunit (β1, δ or ε) was replaced with cDNA coding "empty" pcDNA3.1. For the expression of the human α7 AChR, we used a total of 3.0 µg/dish of cDNAs coding the α7 subunit, RIC-3, and NACHO in a 1:5.5:5.5 ratio (by weight). For the expression of the human α3β4 AChR, we used a total of 3.0 µg/dish of cDNAs coding the α3 subunit, the β4 subunit, and RIC-3 in a 1:1:1 ratio (by weight); some transfections also included cDNA coding NACHO (using the same amount of total cDNA in a 1:1:1:1 ratio). No clear differences in the kinetics of the recorded time courses could be detected between these

two types of transfection. For the expression of the human α1β2γ2L or α1β3γ2L GABA$_A$Rs, we used a total of 3.0 µg/dish of cDNAs coding the α1 subunit, either the β2 or β3 subunit, and the γ2L subunit in a 1:1:4 ratio (by weight). Mutations were engineered using the QuikChange kit (Agilent Technologies), and the sequences of the resulting cDNAs were verified by dideoxy sequencing of the entire coding region (ACGT). For pairwise structural comparisons, atomic models of the same protein in different conformations were globally superposed by minimizing the root mean square deviation (RMSD) of all Cα atoms in all five subunits using VMD[57].

### Electrophysiology

Macroscopic currents were recorded in the whole-cell configuration of the patch-clamp technique at ~22 °C with an effective bandwidth of DC−5 kHz using an Axopatch 200B amplifier (Molecular Devices). Currents were digitized at 10 or 100 kHz and analyzed using pCLAMP 11.1 software (Molecular Devices). Series-resistance compensation was used and set to ~80%. The reference Ag/AgCl wire was connected to the extracellular solution through an agar bridge containing 200 mM KCl. Agonist-concentration jumps were applied to whole cells using a piece of double-barreled θ-tubing (Siskiyou). The flow of solutions through the θ-tube was controlled using a gravity-fed system (ALA BPS-8; ALA Scientific Instruments), and the movement of the θ-tube was achieved using a piezo-electric arm (Burleigh-LSS-3100; discontinued) controlled by pCLAMP 11.1 software (Molecular Devices). The latter signals were low-pass filtered (900 C; Frequency Devices) at a cutoff frequency of ~25−35 Hz prior to their arrival at the piezoelectric arm to reduce ringing in the θ-tube motion. During experiments, patched cells remained attached to a piece of collagen-coated glass coverslip (Neuvitro) placed at the bottom of the recording chamber. In this configuration, the perfusion system achieved a solution-exchange time of ~1.0 ms for the $t_{10-90\%}$ and ~2.0 ms for the $t_{90-10\%}$, as estimated from changes in the liquid-junction current measured with an open-tip patch pipette. Although slower and more variable than the perfusion that can be achieved with excised patches, the whole-cell configuration was favored here so as to increase the number of channels contributing to the observed currents. Whole-cell recordings were performed at a fixed pipette potential (−60 mV; to probe the kinetics of gating) or at a series of closely spaced potentials straddling the channel's reversal potential (to estimate the channel's charge selectivity under dilution conditions). For all recordings from GLIC chimeras, the pipette solution was 110 mM KCl, 40 mM KF, and 5 mM HEPES/KOH, pH 7.4. For recordings from GLIC chimeras performed at a fixed pipette potential, the bath solution was 142 mM NaCl, 5.4 mM KCl, 1.8 mM CaCl$_2$, 1.7 mM MgCl$_2$, 200 µM amiloride (Millipore-Sigma), and 0.1% v/v DMSO (extracellular solution A) adjusted to pH 7.4 with 10 mM HEPES/NaOH; and the two solutions flowing through the barrels of the θ-tube (to which cells were alternately exposed) were bath solution and extracellular solution A adjusted to pH 4.5 with 10 mM acetic acid/NaOH. For recordings from GLIC chimeras aimed at determining reversal potentials, the bath solution was 15 mM KCl, 230 mM mannitol, 200 µM amiloride (Millipore-Sigma), 100 µM DIDS (Millipore-Sigma), and 0.2 % (v/v) DMSO (extracellular solution B) adjusted to pH 7.4 with 5 mM HEPES/KOH; and the two θ-tube solutions were bath solution and extracellular solution B adjusted to pH 4.5 with 5 mM acetic acid/KOH. Both amiloride and DIDS were freshly added to the corresponding solutions from concentrated stocks in DMSO on the day of the experiments. For recordings from the α7 AChR, the α1β1δε AChR, the α3β4 AChR, and the α1β2/3γ2L GABA$_A$Rs, the pipette solution was 110 mM KCl, 40 mM KF, and 5 mM HEPES/KOH, pH 7.4; the bath solution was 142 mM NaCl, 5.4 mM KCl, 1.8 mM CaCl$_2$, 1.7 mM MgCl$_2$, and 10 mM HEPES/NaOH, pH 7.4; and the two θ-tube solutions were bath solution with or without agonist. The agonist was 100 µM ACh for the α7 AChR and the α1β1δε AChR; 200 µM ACh for the α3β4 AChR; and 1 mM GABA for the α1β2/3γ2L GABA$_A$Rs. Patch pipettes, pulled

from thin-walled borosilicate-glass capillary tubing (Sutter Instrument), had resistances of 3−5 MΩ when filled with pipette solution.

Single-channel currents from the α1β1δε AChR (wild-type and mutants) were recorded in the cell-attached configuration at −100 mV (pipette potential) and ~22 °C with an effective bandwidth of DC−30 kHz using an Axopatch 200B amplifier (Molecular Devices). Currents were digitized at 100 kHz and analyzed using QuB 1.4 software (The MLab Edition[58]). The bath solution was 142 mM NaCl, 5.4 mM KCl, 1.8 mM CaCl$_2$, 1.7 mM MgCl$_2$, and 10 mM HEPES/NaOH, pH 7.4; and the pipette solution was 110 mM KCl, 40 mM KF, and 1 µM ACh adjusted to either pH 7.4 with 10 mM HEPES/KOH or pH 6.0 with 10 mM MES/KOH. Patch pipettes, pulled from thick-walled borosilicate-glass capillary tubing (Sutter Instrument), had resistances of 7−10 MΩ when filled with pipette solution.

### Plasma-membrane expression

The number of muscle AChRs on the plasma membrane of transfected HEK-293 cells was estimated from the amount of [$^{125}$I]-α-bungarotoxin ([$^{125}$I]-α-BgTx; Revvity) bound upon incubation with a saturating concentration of toxin (~30 nM)[1,2]. α-BgTx is a (membrane-impermeant) 74-amino-acid snake neurotoxin that binds to the neurotransmitter-binding sites of the muscle AChR. 24 h after the transfections were terminated, transfected cells were resuspended in a HEPES-buffered sodium-saline solution (142 mM NaCl, 5.4 mM KCl, 1.8 mM CaCl$_2$, 1.7 mM MgCl$_2$, and 10 mM HEPES/NaOH, pH 7.4) by gentle agitation, and [$^{125}$I]-α-BgTx was added. Toxin binding-reaction mixtures (1mL each in 1.7-mL plastic tubes) were incubated with constant rotation for 4 h at 4°C, and upon completion, cell-bound label was separated from unbound label by centrifugation at 16,000 × g for 3 min at room temperature. To reduce the amount of label bound non-specifically to the cells, these pellets were resuspended in 1mL of ice-cold Dulbecco's phosphate-buffered saline (pH 7.4; Gibco), vortexed for 30 s, and pelleted again at 16,000 × g for 3 min at room temperature; this resuspension−pelleting procedure was repeated twice. Washed pellets were, then, resuspended in a solution containing 0.1 N NaOH and 1% (w/v) sodium dodecyl sulfate (SDS), and incubated at 65−70°C for 30 min. The radioactivity and protein content of each solubilized pellet were estimated: $^{125}$I radioactivity was measured for 1 min using a Wiper$^{TM}$ 100 γ-counter (Laboratory Technologies, Inc.), and the amount of protein was estimated using the bicinchoninic-acid assay (BCA; ThermoFisher) and a freshly prepared bovine serum-albumin (ThermoFisher) calibration curve. As a control of the non-specific binding of [$^{125}$I]-α-BgTx to transfected cells, HEK-293 cells were transiently transfected with cDNAs coding the mouse β1, δ, and ε (but not α1) AChR subunits. These cells were incubated with [$^{125}$I]-α-BgTx under the same conditions as were the cells transfected with all four types of muscle-AChR subunits.

### Reporting summary

Further information on research design is available in the Nature Portfolio Reporting Summary linked to this article.

## Data availability

All data are available in the main text or the Supplementary Information. PDB codes of previously published structures used in this study are: 6HUG; 6HUO; 7KOQ; 7KOX; 7EKI; 8I42; 8WCR; 7QKO; and 7QL5. The source data underlying Figs. 5, 6, 7b,c, 8, and 9 can be found in the Source Data file. Source data are provided with this paper.

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

## Acknowledgements

We thank Nicole E. Godellas and Lauren Lane for assistance. We also thank William N. Green, Ciria C. Hernandez, Robert L. Macdonald and Steven M. Sine for providing cDNAs. This work was supported by the School of Molecular and Cellular Biology of the University of Illinois Urbana-Champaign (C.G.) and a grant from the US National Institutes of Health (R01-NS042169, C.G.).

## Author contributions

Conceptualization: G.D.C. and C.G. Formal Analysis: G.D.C. and C.G. Investigation: G.D.C. Supervision: C.G. Writing—original draft: C.G. Writing—review & editing: G.D.C. and C.G.

## Competing interests

The authors declare no competing interests.
