## [Transparent Peer Review file · Nature Communications]

Disentangling the mechanistic role of loop-C capping in Cys-loop receptor activation

Corresponding Author: Dr Claudio Grosman

Version 0:

Reviewer comments:

Reviewer #1

(Remarks to the Author)

This study shows that deletion of loop C in GLIC chimeras or in the subunits of the muscle nAChR that contain an allosteric loop C do not eliminate or significantly change channel activation/opening and desensitization. The authors conclude that capping of loop C is not necessary for communication of agonist binding to the TMD to produce opening and desensitization. Instead, Loop C capping may be related to agonist binding to the low-affinity conformation and/or transition in the agonist binding site from the low to high-affinity conformation. While the findings in the GLIC chimeras are informative about the role of Loop C in a GLIC ECD, it is the reviewer's opinion that the study's conclusion as stated does not unequivocally follow from the results. An alternative explanation to the results in the GLIC chimera could be that proton activation bypasses the loop C rearrangement because of the distinct locations of the proton sensors. Protons likely produce rearrangements in other regions of the GLIC ECD that are sufficient to produce channel gating. Indeed, the authors previously showed that deletion of loop C in GLIC does not prevent proton activation. One could liken this to a mutation in the TMD or ECD-TMD interface that renders a channel spontaneously open. Such a channel might still be able to open spontaneously if Loop C was deleted since opening is no longer dependent on agonist binding to the orthosteric site. Based on this argument, the data does not rule out the possibility that loop C rearrangement is a necessary mechanical link for the communication of agonist binding to channel opening in pLGICs that depend on agonist binding to the orthosteric site.

The manuscript is clearly written but the figures use structures from multiple channels without any clear reasoning. Figures 1 and 2 show the alpha7 nAChR, Fig. 4 shows the GABA(A)R, and the chimeric constructs use GluCl and GlyR. It may be easier for the reader to interpret these structures and the argument being made from these figures if there was more consistency in the structures used or a reason expressed in the text for each chosen structure.

A few additional minor points:

- 1) Regarding the incorporation of loop C deleted subunits in the muscle nAChR, is it possible that in the macroscopic recordings, there is a mixture of receptors with the loop C deleted subunit incorporated and omitted? In the single channel recordings, how many channels/recordings were obtained and were there any recordings that showed WT characteristics even with expression of the Loop C-deleted subunit? It is clear that the Loop C deleted subunit are expressed in at least some of the channels so the results are still convincing. Nevertheless, it would be helpful to comment on this.
- 2) Page 6 line 146 states that 50 residues are lacking from the ECD-TMD covalent junction. Do the 50 residues refer to the 10 x 5 residues in Loop C? If so, these are not at the ECD-TMD junction.
- 3) Page 7 line 188 refers to 9' and Fig. 1 but this is not indicated in Fig. 1.

Reviewer #2

(Remarks to the Author)

please see attached file

Reviewer #3

(Remarks to the Author)

The central question of the study is “whether the pore of agonist-bound Cys-loop receptors requires the capping of loop C to gate”. To address this, the authors make a very clever use of chimeras (between the extracellular domains (ECDs) of the proton-gated prokaryotic GLIC channel and the transmembrane region (TM) of animal Cys-loop receptors) as well as heteromeric AChRs with some of the C- loops deleted and a reporter-mutation that confers proton-sensitivity (L9'H). They obtained beautiful macroscopic and single-channel currents from well expressing chimeras clearly showing channel kinetics comparable to wt-like constructs. Their conclusion is, thus, sound: “capping rearrangement of loop C is not required for the pore of small-molecule-gated Cys-loop receptors to open/desensitize in response to agonist binding”.

The manuscript is very well written, and the results bear significance for the activation pathway of the whole Cys-loop receptor superfamily whose relevance in human neurophysiology cannot be overstated.

My main comment would be that the work could benefit from more direct explanations of what is already known about the role of loop C rearrangements in activation of Cys loop receptors to make the novelty clearer. For example, it is well documented that AChRs can gate unliganded (Purohit, PNAS, 2009), when all or some C loop residues are mutated into glycines (Purohit, JGP, 2013) and when C loop is apparently trapped in a capped state via disulphide bridges (Muhktasimova, Nature, 2009). So, it is clear that, at least in AChRs, loop C in general is not needed for pore opening and that ligand binding followed by loop C capping is more involved in shifting the equilibrium between closed and open ion channel. The authors are very careful and consistent throughout the manuscript to explain that they interested into whether loop C capping is necessary for pore opening in agonist-bound receptors. They are not investigating whether loop C in general is needed for gating, which is what the studies above are addressing (the authors do cite Purohit, JGP, 2013 study, but I am not sure about the other two). Ideally then, to see whether loop C capping is needed for channel gating in agonist-bound receptors, one would need to somehow immobilize loop C while still allowing agonists to bind (basically, the opposite cross-link from the one used in the Muhktasimova study). Proton-gated GLICs are not ideal in a sense that here, again, we end up deleting C loops while the agonists are bound elsewhere on the protein, not allowing us to address directly the capping movement in agonist-bound orthosteric sites. Such cross-linking might not be possible and therefore, using proton-gated GLICs might be the best we can do experimentally (I guess additional approach could be MDs?). Also, for all we know, AChRs where 9 loop C residues have been mutated into Gly and 2 aromatic residues left unchanged and which can still bind Ach might have a decreased or no capping capacity (Purohit, JGP, 2013) due to extensive mutations. I am sure the authors can do a much better job discussing the literature mentioned above, but my point is that stating what is already out there in terms of loop C mutants would make it easier for the reader to understand the specificity of the question/phenomenon being studied here.

I do like the way the authors are showing the current traces, but why not quantify the patches and show the values in a table, for example, so that the reader can see “...these kinetics are comparable to those displayed by naturally occurring members of the superfamily (under similar experimental conditions), a finding that lends further credence to the use of the chimeric approach reported here” (rows 147-149).

Some minor comments:

- 1) I am not sure differences between GLICs and animal Cys loop receptors were noted anywhere – it would be good to include this.
- 2) Figs. 3 and 4 are nice, but maybe include alignment of GLIC and GABA receptors with a capped loop C? Or something that will allow us to compare the two structures directly.
- 3) Fig. 3 – maybe show where the major binding sites for protons are?
- 4) Rows 107-108, the authors write “We reasoned that, if the capping of loop C were required for pore gating, then deleting this loop would lead to a total loss-of-function phenotype”, but we already know that, at least for some Cys loop receptors, such as AChRs, loop C capping is not needed for gating (see above). So, is this really the right thing to reason here?
- 5) Row 129-130: “Fast-perfusion patch-clamp experiments revealed that the GLIC-β-GluCl chimera generated the largest currents, followed by the GLIC-α1-GlyR construct.” – were the currents with other constructs just small or not present at all? If they were present, but small in amplitude, it might be worth still including those data to show the generality of the approach for the whole Cys loop receptor superfamily.
- 6) Row 188 – identify position 9' in Fig. 1.
- 7) Maybe use AlphaFold to check the structure of the GLIC-beta-GluCl chimera and to show its similarity to wt Cys loop receptors?

Version 1:

Reviewer comments:

Reviewer #1

(Remarks to the Author)

I am satisfied with the response and revisions by the authors to my comments as well as those of the other reviewers. The data support the author's conclusions and claims. I have no further comments or requests for revisions.

Reviewer #2

(Remarks to the Author)

The authors have fully addressed the comments of all reviewers from the initial round of review in May.

The authors use deletion constructs and electrophysiology in the bacterial pentameric channel GLIC to investigate the relationship between loop C closure and channel opening. Based on prior observations loop C closure has been considered required for channel opening. Loop C caps a cavity that can close on a bound agonist. The authors here show that loop C is indeed not required for channel opening, but rather other potentially subtle changes initiate the transition from closed to open channels.

Reviewer #3

(Remarks to the Author)

Thank you for including the discussion about the role of loop C in unliganded gating. The discrepancies in the published literature, as pointed out by the authors, are very interesting and probably not that well known outside the field. I think the mention of this gives a better and more nuanced context to their work.

“However, because the structural aspects of unliganded gating of Cys-loop receptors are poorly understood, it is still unclear how the ECD and TMD communicate in the complete absence of bound ligands.” – Very true and I am not saying that the authors need to mention this anywhere, but just as a side note: it is possible that unliganded gating has a very different gating mechanism to the liganded gating, but also that Cys-loop receptors can transition from closed to open states without a ligand and without the loop C capping, but the binding of a ligand and the subsequent loop C capping shift this equilibrium, with the underlying mechanism being the same. I would even argue that similarities between GLIC, which is not activated by small agonists, and other members of the superfamily (as nicely explained by the authors in rows 114-122) suggest this might be the case.

I appreciate, in particular, the inclusion of additional data in Figs. 5c and 6c. I think the new traces, together with the clearly stated number of traces and cells, clearly illustrate how comparable the kinetics across Cys-loop superfamily is.

I take the authors' point on comparing the structures from different receptors and on AlphaFold and thank them for including the protonation sites on GLIC in the supplementary material – I found this helpful.

Point-by-point response to Reviewer #1

We thank Reviewer #1 for their careful reading of our manuscript and for the important suggestions they have made. In the new document, all new or re-written passages of text are highlighted with a cyan background. Below, Reviewer #1's comments are *italicized* and in blue font, and text taken from the manuscript is underlined.

1. While the findings in the GLIC chimeras are informative about the role of Loop C in a GLIC ECD, it is the reviewer's opinion that the study's conclusion as stated does not unequivocally follow from the results.

All instances of the words “unequivocally” and “unambiguously” were eliminated.

2. An alternative explanation to the results in the GLIC chimera could be that proton activation bypasses the loop C rearrangement because of the distinct locations of the proton sensors. Protons likely produce rearrangements in other regions of the GLIC ECD that are sufficient to produce channel gating. ... Based on this argument, the data does not rule out the possibility that loop C rearrangement is a necessary mechanical link for the communication of agonist binding to channel opening in pLGICs that depend on agonist binding to the orthosteric site.

As we explicitly state in a new paragraph of this revised version (p.9, ln 245–253), our approach is necessarily indirect. Clearly (p.4, ln 90–94), the most compelling way to tackle this question would be to delete the loops C of a Cys-loop receptor and probe for function on application of agonists. However, because these loops ... are necessary for binding agonists in the first place, agonist-activated currents would not be elicited from such a construct, and therefore, the question could not be answered. To emphasize this point from the very outset, we added a new sentence to the Abstract (p.2, ln 27–28): However, because binding and gating are inextricably linked, testing this idea experimentally has proved challenging. Hence, if we agree that the question regarding the role of loop-C capping in channel activation needs to be addressed, then we also need to agree to resort to experimental approaches that are indirect. Importantly, however, the observation that the structural rearrangements at the ECD–TMD interface are conserved—whether protons bind to a fuzzy array of ionizable side chains or small-molecule agonists occupy well-defined pockets (see also ref. 47 for a similar conclusion)—mitigates concerns about the indirect nature of our approach. Therefore, the end of this new paragraph (p.9, ln 251–253) reads: ... although the (necessarily) indirect nature of our experimental approach does not allow us to rule it out completely, the possibility that our conclusions pertain only to proton-gated constructs of the superfamily seems also very unlikely.

3. The manuscript is clearly written but the figures use structures from multiple channels without any clear reasoning. Figures 1 and 2 show the alpha7 nAChR, Fig. 4 shows the GABA(A)R, and the chimeric constructs use GluCl and GlyR. It may be easier for the reader to interpret these structures and the argument being made from these figures if there was more consistency in the structures used or a reason expressed in the text for each chosen structure.

The use of different members of the superfamily in experiments (β GluCl and α 1 GlyR) and figures (α 7 AChR in Figs. 1c and 2 and α 1 β 3 γ 2L GABA_AR in Fig. 4) was deliberate; we wanted to include a variety of pLGICs. Moreover, in response to a comment by Reviewer #2, we have now added two figure panels (Figs. 5c and 6c) in which the deactivation time courses of the chimeras are compared to those of several wild-type AChRs (α 7, α 3 β 4, and α 1 β 1 δ ϵ) and GABA_ARs (α 1 β 2 γ 2L and α 1 β 3 γ 2L). Indeed, the entire idea behind this manuscript is that domain–domain communication seems to proceed through a largely conserved structural mechanism. Hence, there was no particular rationale behind the choice of the α 7 AChR for Figures 1 and 2; we simply wanted to show a homomeric Cys-loop receptor, and we had to choose one. Similar is the case for Figure 4 and the α 1 β 3 γ 2L GABA_AR; we wanted to show a heteromeric channel and, because the α 7 AChR of Figures 1 and 2 is cation selective, we chose an anion-selective member, instead. To clarify these choices, new text was added in p.3 ln 66–67: Figure 2 illustrates this phenomenon with atomic models of the α 7-AChR and p.4 ln 110–111: Figure 4 illustrates the latter rearrangements using a heteromeric GABA_AR as an example. Furthermore, both figure legends have now new text that reinforces the idea that the α 7 AChR and the α 1 β 3 γ 2L GABA_AR are being used as mere examples. Regarding the new Figures 5c and 6c, once again, we had to choose some small-molecule-gated Cys-loop receptors to perform a comparison—there was no particular rationale behind these choices other than the intention of covering representative examples of the major pLGIC types.

A few additional minor points:

1) Regarding the incorporation of loop C deleted subunits in the muscle nAChR, is it possible that in the macroscopic recordings, there is a mixture of receptors with the loop C deleted subunit incorporated and omitted? In the single channel recordings, how many channels/recordings were obtained and were there any recordings that showed WT characteristics even with expression of the Loop C-deleted subunit? It is clear that the Loop C deleted subunit are expressed in at least some of the channels so the results are still convincing. Nevertheless, it would be helpful to comment on this.

In addition to creating a proton-binding site clearly identifiable in single-channel recordings, the reporter mutation (L9'H) slows down the macroscopic-current decay time courses, which makes it easier to estimate the contribution of channels carrying the mutant subunit to the entire population of expressed receptors. To explicitly emphasize this point, we added the following text (p.8, ln 208–216): Furthermore, the slower kinetics of macroscopic-current decay recorded from the mutants (Fig. 7b, c) indicate that the loop-C deleted + L9'H subunits express well and are present in a sizable fraction—perhaps, all—of the plasma-membrane AChRs. Evidently, we cannot rule out the possibility that some subunit-omitted receptors expressed and contributed to the macroscopic currents. However, since currents through subunit-omitted muscle AChRs do not decay more slowly (Fig. 8), these aberrant AChRs could at most account for only a small fraction of the expressed AChRs. It follows then that any single allosteric loop C is largely dispensable for expression, assembly, and function of the muscle AChR.

Regarding the single-channel recordings, these were performed in the presence of 1- μ M

ACh, and at this concentration, bursts of openings from subunit-omitted muscle AChRs are very brief. Although brief events were clearly seen in recordings from loop-C-deleted + L9'H muscle AChRs, these events are so short-lived that inspecting their pH-dependent subconductance behavior (so as to classify them as bursts from a subunit-omitted muscle AChR versus as very-short-lived bursts from a loop-C-deleted + L9'H muscle AChR) was not possible. Recordings from L9'H mutants of the muscle AChR—without the loop-C deletion—also typically contain brief bursts of openings.

We hope that the added text (p.8, ln 208–216) suffices to clarify this point. We are afraid that further elaboration on the types of bursts observed in single-channel recordings will distract from the main idea, namely, that a heteromeric Cys-loop receptor (here, the muscle AChR) can gate even if its allosteric loops C are deleted.

2) Page 6 line 146 states that 50 residues are lacking from the ECD-TMD covalent junction. Do the 50 residues refer to the 10 x 5 residues in Loop C? If so, these are not at the ECD-TMD junction.

Yes, we are referring to the 10×5 (= 50) residues deleted from the five loops C, and we had written that they are near (not, at) the junction. To make the proximity of the deleted residues to the covalent junction clearer, this part of the revised version (p. 6, ln 152–154) now reads: ... which is remarkable considering that the mutant lacks 10 residues per subunit (for a total of 50 residues) only 12 residues away from the ECD–TMD covalent junction (Fig. 3a).

3) Page 7 line 188 refers to 9' and Fig. 1 but this is not indicated in Fig. 1.

Thank you very much for catching this mistake. This sentence (p.7, ln 193–196) now refers the reader to Figure 1a only to see the location of the pore-lining α -helix M2 (but not the 9' position). We explicitly indicate that position 9' is in the middle of the pore-lining α -helix M2 (p.7 ln. 195–196), and thus, we think that adding one more caption to Figure 1 is unnecessary.

Point-by-point response to Reviewer #2

We thank Reviewer #2 for reading our manuscript and for their comments. In the new document, all new or re-written passages of text are highlighted with a cyan background. Below, Reviewer #2's comments are *italicized* and in blue font, and text taken from the manuscript is underlined.

The work is original and to the best of my knowledge C-loop deletions have not been published for pentameric channels.

We are pleased to learn Reviewer #2 deemed our work to be original.

The established literature, also cited by the authors, shows that using structure determination (mostly cryo-EM) that loop C closing behavior is not the same across all pentameric channel subunits.

Indeed. Differences in the extent of loop-C capping among Cys-loop receptors are explicitly elaborated in the second paragraph of the Introduction, and differences among the subunits of heteromeric Cys-loop receptors are noted on p.7, ln 172–173.

PTX – GABA structure shows loop C closure in beta subunits ... the channels are closed in that condition due to PTX functioning as a channel blocker. This structure would show that the loop can close on an agonist in the absence of opening since the channel conformational changes are blocked by the channel blocker and also block ECD concerted constricting conformational changes.

Here, Reviewer #2 may be referring to an atomic model of the $\alpha 1\beta 3\gamma 2L$ GABA_AR bound to both GABA and pore blocker PTX (PDB ID 6HUJ). In this structure, the “orthosteric” loops C are capped, yet the PTX-bound pore appears to be closed. Because pore blockade of GABA_ARs by PTX is known to interfere with the channel's closed \rightleftharpoons open \rightleftharpoons desensitized conformational interconversions (e.g., PMID: 20487876 and PMID: 6308160)—and has recently been proposed to uncouple the extracellular and transmembrane domains (PMID: 37659407)—we decided not to include structures of PTX-bound GABA_ARs in our Figure 4. After all, our manuscript focuses on the involvement of loop-C capping in domain–domain coupling. Hence, the conformational rearrangements at the level of the ECD–TMD interface upon the binding of GABA are illustrated using atomic model 6HUO, which was obtained in the absence of PTX.

Benzodiazepines and GABA structures show a loop C (alpha subunit) slight opening, benzos promote channel opening. This structure would show that opening of loop C leads to improved channel opening.

We surmise Reviewer #2 is referring here to atomic model 6HUO (or to comparable models of $\beta 1$ - or $\beta 2$ -subunit containing $\alpha\beta\gamma$ -type GABA_ARs), which shows an uncapped benzodiazepine-bound loop C and a desensitized pore. Thus, it could be argued that this observation alone

answers the question as to whether loop-C capping is required for the pore of Cys-loop receptors to open/desensitize in response to agonist binding to the extracellular domain. However, it should also be pointed out that: *i*) In addition to the single-molecule of bound benzodiazepine (alprazolam), this model also features two molecules of bound GABA; *ii*) The two GABA-binding loops C are fully capped; and *iii*) The binding of alprazolam alone does not open/desensitize the wild-type channel. Therefore, we think that the mere observation of atomic models does not allow us to answer the questions posed in our manuscript. Instead, it seems clear to us that experiments that probe receptor-channel function are required.

Several of the amino acids deleted in the GLIC loop-C deletions here are shown to be involved in proton sensing. However, there are many residues all throughout GLIC that contribute to proton sensing and channel opening For GLIC proton sites have been described all throughout the ECD and TMD. No single mutation or combination so far has been shown to eliminate proton gating. Therefore, GLIC opening does per se not require an agonist binding to loop C, even though loop C does contain proton sensing involved residues.

Certainly. The protonation of residues in regions other than loop C suffices to activate the channel. To make this idea clearer, we added a new figure (Supplementary Fig. 1) that shows the location of all ionizable residues in the extracellular domain of GLIC.

Since we can reasonably well assume that protons are binding to multiple sites within the ECD,

Actually, we don't need to make assumptions; it has been experimentally shown that the protons that drive gating of GLIC bind to multiple side chains within the ECD (*e.g.*, refs. 28, 29). The fact that GLIC gates upon binding ligands (protons) to ECD residues other than those in loop C is the very reason why we chose this channel to study the effects of loop-C deletion on gating of Cys-loop receptors.

... they can be assumed to change the conformation of the ECD which in turn changes the conformation of the TMD.

To avoid making assumptions, we decided to test these ideas experimentally.

note: deleting all three other subunit loop C at the same time led to non-functional channels. This makes us think that overall loop C is important for function in heteropentameric eukaryotic channels.

We would like to clarify this point. As we wrote in the Results section (p. 7, ln 182–184), Deleting all three allosteric loops C prevented the expression of the receptor on the plasma membrane, as revealed by the absence of specific binding of membrane-impermeant [¹²⁵I]- α -bungarotoxin to the cell surface of transfected HEK-293 cells. In other words, deleting the loops C of the β 1, δ , and ϵ subunits of the muscle AChR (all three) reduced cell-surface expression to undetectable levels, and hence, the effect of this triple loop-C deletion on function could not be assessed. Hence, we do not know whether deleting all three loops C leads to non-functional

channels; the triple mutant, simply, does not express.

Appreciate the effort in demonstrating that for the channels recorded in the single channel patches there is a titratable H at 9' demonstrating that those individual channels do have the C loop deleted subunit; however, this does not mean that in the whole cell currents all or most channels do have the subunits incorporated.

In addition to creating a proton-binding site clearly identifiable in single-channel recordings, the reporter mutation (L9'H) slows down the macroscopic-current decay time courses, which makes it easier to estimate the contribution of channels carrying the mutant subunit to the entire population of expressed receptors. To explicitly emphasize this point, we added the following text (p.8, ln 208–216): Furthermore, the slower kinetics of macroscopic-current decay recorded from the mutants (Fig. 7b, c) indicate that the loop-C deleted + L9'H subunits express well and are present in a sizable fraction—perhaps, all—of the plasma-membrane AChRs. Evidently, we cannot rule out the possibility that some subunit-omitted receptors expressed and contributed to the macroscopic currents. However, since currents through subunit-omitted muscle AChRs do not decay more slowly (Fig. 8), these aberrant AChRs could at most account for only a small fraction of the expressed AChRs. It follows then that any single allosteric loop C is largely dispensable for expression, assembly, and function of the muscle AChR. Importantly, regardless of the extent to which the loop-C deleted subunits are incorporated into the assembled AChRs, the single-channel recordings in Figure 9 provide compelling support to our conclusion: heteromeric Cys-loop receptors (here, the muscle AChR) can gate even if its allosteric loops C are deleted.

We are not expecting the beta, delta or epsilon C loops to close as they do not harbor agonist sites. So why does it matter that their C loop deleted counterparts (if they are quantitatively incorporated in the pentamers) are still functional. It would be the expectation that they would be functional.

We would rather not predict the functional relevance of a structural element of a protein simply on the basis of the extent of its rearrangement between interconverting conformations. As we wrote in the Results section (p. 7, ln 172–179), Although the extent to which the allosteric loops C rearrange upon agonist-binding and gating seems to be very small^{10–13} (Fig. 7), a role in domain–domain coupling could be expected from every single loop C inasmuch as the low-to-high affinity change of the ECD needs to be transmitted to all five subunits of the TMD for the pore to open and desensitize. Moreover, studies of neurotoxins that bind to the allosteric loops C of neuronal AChRs⁴⁰ or GABA_ARs⁴¹ revealed that toxin binding has clear effects on agonist-evoked currents, further hinting at the possibility of a role for these loops in the communication between the ECD and the TMD. In conclusion, we think that there are good reasons to investigate the role of allosteric loops C in liganded gating.

Point-by-point response to Reviewer #3

We very much appreciate Reviewer #3's encouraging comments and most insightful suggestions. In the new document, all new or re-written passages of text are highlighted with a cyan background. Below, Reviewer #3's comments are *italicized* and in blue font, and text taken from the manuscript is underlined.

1) My main comment would be that the work could benefit from more direct explanations of what is already known about the role of loop C rearrangements in activation of Cys loop receptors to make the novelty clearer. For example, it is well documented that AChRs can gate unliganded ...when all or some C loop residues are mutated into glycines (Purohit, JGP, 2013) and when C loop is apparently trapped in a capped state via disulphide bridges (Muhktasimova, Nature, 2009)... So, it is clear that, at least in AChRs, loop C in general is not needed for pore opening I am sure the authors can do a much better job discussing the literature mentioned above

The paper by Purohit and Auerbach (2013) (our reference 23) dealt with unliganded gating of the muscle AChR. The authors found that, although extensive mutagenesis of the loops C of the $\alpha 1$ subunits (that is, the "orthosteric" loops C) prevented activation by ACh, these mutations had, essentially, no major effect on unliganded gating. From this result, and making the crucial assumption that the mechanism underlying ECD-TMD coupling in unliganded and liganded receptors is the same, Purohit and Auerbach extended these observations and proposed that loop-C capping is not required for liganded gating, either. However, because the structural aspects of unliganded gating of Cys-loop receptors are poorly understood, it is still unclear how the ECD and TMD communicate in the complete absence of bound ligands. Thus, to address the question as to whether the capping of loop C is the mechanical link that couples the ECD to the TMD during AGONIST-DRIVEN gating, we resorted to the experimental approach described in our manuscript.

Reviewer #3 also mentions the 2009 paper by Nuriya Muhktasimova, Steven Sine, and coworkers (PMID: 19339970), which also dealt with unliganded gating of the muscle AChR. In this paper, the oxidation of a pair of strategically engineered cysteines per orthosteric site was proposed to lock loop C in the capped position. According to these authors, this is a conformation ("primed conformation") that needs to be attained in order for the unliganded muscle AChR to gate. Certainly, Sine and coworkers proposed that, when both orthosteric loops C are capped, unliganded openings are long; when only one is capped, unliganded openings are short; and when neither loop C is capped, no openings occur (page 453 of their paper). Hence, it follows that loop-C capping would be required for unliganded gating, a conclusion that is difficult to reconcile with that in Purohit and Auerbach (2013).

To summarize, using similar methodologies (mutagenesis and single-channel electrophysiology) on the very same Cys-loop receptor, these two papers led to different conclusions as to the requirement of loop-C capping for the unliganded channel to open. Because we are not dealing with unliganded gating in our manuscript, we cannot shed any light on this (perhaps, little known) controversy. Hence, we would like to cite these two papers only

briefly while explicitly noting the lack of consensus on this matter (p. 4, ln 94–99): It is not surprising, then, that the role of loop-C capping in the binding–gating coupling of Cys-loop receptors has remained difficult to ascertain (e.g.,^{22–26}). In fact, to our knowledge, the effects of loop-C perturbations on gating—without the confounding effects of altered ligand binding—have thus far only been studied by omitting the ligand altogether. Unfortunately, however, published reports on the requirement of loop-C capping for unliganded gating have been conflicting^{22,23,26}. Reference 22 is Mukhtasimova et al. (2009), reference 23 is Purohit and Auerbach (2013), and reference 26 is a more recent (2024) reference by Auerbach and coworkers on the topic of loop-C capping using a computational approach. References 24 and 25 are papers by other authors who noted the discrepancies in the literature regarding the functional role of loop-C capping.

Reviewer #3 also proposes that we cite a 2009 PNAS paper by Purohit and Auerbach. Because this is a rather technical paper on the estimation of rate and equilibrium constants of unliganded gating, we find it unnecessary to include it among our references; we hope Reviewer #3 finds this to be acceptable.

2) I do like the way the authors are showing the current traces, but why not quantify the patches and show the values in a table, for example, so that the reader can see “...these kinetics are comparable to those displayed by naturally occurring members of the superfamily ...” (rows 147-149).

We thank this reviewer for this excellent point, which we took very seriously. To increase the statistical significance of our statement about “comparable kinetics”, we set out to record a large number of individual current responses to agonist applications from cells expressing five different naturally occurring Cys-loop receptors (a homomeric AChR, two heteromeric AChRs, and two heteromeric GABA_ARs). We found, however, that showing clearly the similar kinetics of the chimeras and these different constructs with numbers (say, with a table of time constant and amplitudes of exponential fits) was not straightforward. This is because current-decay time courses were often best fit with more than a single exponential component, and the best number of components differed among constructs. Instead, what we decided to plot in the new Figures 5c and 6c is the chimeras’ average responses (GLIC–β-GluCl in Fig. 5c and GLIC–α1-GlyR in Fig. 6c; the same black traces as in the respective panels a) along with the average responses of the different naturally occurring Cys-loop receptors we selected (the same five for Figs. 5c and 6c). Although less quantitative than a table with numbers, we feel that this type of display is very effective in conveying the idea of “comparable kinetics”. Insets in Figures 5c and 6c indicate the number of responses that were averaged for each construct and the number of cells from which the data were obtained.

Some minor comments:

1) I am not sure differences between GLICs and animal Cys loop receptors were noted anywhere – it would be good to include this.

The legend to Figure 3 now reads (toward the end): Compared to small-molecule gated Cys-

loop receptors from animals, GLIC has a much shorter M3–M4 linker. In our opinion, this is the most obvious difference.

2) Figs. 3 and 4 are nice, but maybe include alignment of GLIC and GABA receptors with a capped loop C? Or something that will allow us to compare the two structures directly.

The differences between the unliganded closed and agonist-bound open or desensitized structures are best appreciated within a given receptor type as illustrated in Figures 3 (for GLIC) and 4 (for a GABA_AR) where the pairwise comparisons are made between structures of the same channel. The superposition of an atomic model of loop-C capped GLIC to an atomic model of a loop-C capped GABA_AR, while possible (see below), seems much less informative because their respective loop-C uncapped models do not superpose perfectly. The example below shows the superposition of a GABA-bound model of the $\alpha 1\beta 3\gamma 2L$ GABA_AR to a model of GLIC at pH 4.0. The superposition was performed using the SSM (Secondary Structure Matching) algorithm in Coot/CCP4. It seems to us that not much can be appreciated from this type of comparison, which is why we are not showing it in the manuscript.

3) Fig. 3 – maybe show where the major binding sites for protons are?

Sure. A new figure (Supplementary Fig. 1) now shows all ionizable residues in the ECD of GLIC. As the legend to this figure reads, Although the protonation of some of these residues was deemed to be more relevant for gating than the protonation of others²⁻⁴, it is still unclear whether liganded gating in GLIC can be ascribed to the protonation of only a few side chains.

4) Rows 107-108, the authors write “We reasoned that, if the capping of loop C were required for pore gating, then deleting this loop would lead to a total loss-of-function phenotype”, but we

already know that, at least for some Cys loop receptors, such as AChRs, loop C capping is not needed for gating (see above). So, is this really the right thing to reason here?

To our knowledge, before our work, the effects of loop-C mutagenesis/cross-linking on gating—without the confounding effects of these mutations on ligand binding—had only been studied by omitting the ligand altogether. And, as mentioned above, even for the same Cys-loop receptor, the literature presents conflicting results regarding the requirement of loop-C capping for unliganded gating to occur. In our manuscript, on the other hand, we are exclusively concerned with gating driven by ligands that bind to the ECD.

5) Row 129-130: “Fast-perfusion patch-clamp experiments revealed that the GLIC-β-GluCl chimera generated the largest currents, followed by the GLIC-α1-GlyR construct.” – were the currents with other constructs just small or not present at all? If they were present, but small in amplitude, it might be worth still including those data to show the generality of the approach for the whole Cys loop receptor superfamily.

Clear currents were only present for the GLIC-β-GluCl and GLIC-α1-GlyR chimeras.

6) Row 188 – identify position 9' in Fig. 1.

Thank you very much for catching this mistake. This sentence (p.7, ln 193–196) now refers the reader to Figure 1a only to see the location of the pore-lining α -helix M2 (but not the 9' position). We explicitly indicate that position 9' is in the middle of the pore-lining α -helix M2 (p.7 ln. 195–196), and thus, we think that adding one more caption to Figure 1 is unnecessary.

7) Maybe use AlphaFold to check the structure of the GLIC-beta-GluCl chimera and to show its similarity to wt Cys loop receptors?

AlphaFold indeed predicts Cys-loop receptor-like structures for the GLIC-β-GluCl and GLIC-α1-GlyR chimeras. However, we wonder how relevant these predictions are especially since AlphaFold also predicts similar structures for the other five GLIC chimeras that we tested here and did not express functionally active channels. Perhaps, the best available indication that the structures of both GLIC-β-GluCl and GLIC-α1-GlyR are unlikely to differ greatly from those of naturally occurring Cys-loop receptors is the observation that both constructs behave electrophysiologically as genuine members of the superfamily, as now explicitly shown in Figures 5c and 6c.

Responses to Reviewers #1–3

Below, reviewers' comments are italicized and in blue font.

Response to Reviewer #1

I am satisfied with the response and revisions by the authors to my comments as well as those of the other reviewers. The data support the author's conclusions and claims. I have no further comments or requests for revisions.

We thank Reviewer #1 very much. The changes we've introduced to the manuscript in response to their comments have noticeably improved it.

Response to Reviewer #2

The authors have fully addressed the comments of all reviewers from the initial round of review in May.

The authors use deletion constructs and electrophysiology in the bacterial pentameric channel GLIC to investigate the relationship between loop C closure and channel opening. Based on prior observations loop C closure has been considered required for channel opening. Loop C caps a cavity that can close on a bound agonist. The authors here show that loop C is indeed not required for channel opening, but rather other potentially subtle changes initiate the transition from closed to open channels.

We thank Reviewer #2 for their detailed reading of our submission and their insightful suggestions.

Response to Reviewer #3

Thank you for including the discussion about the role of loop C in unliganded gating. The discrepancies in the published literature, as pointed out by the authors, are very interesting and probably not that well known outside the field. I think the mention of this gives a better and more nuanced context to their work.

We fully agree.

"However, because the structural aspects of unliganded gating of Cys-loop receptors are poorly understood, it is still unclear how the ECD and TMD communicate in the complete absence of bound ligands." – Very true and I am not saying that the authors need to mention this anywhere, but just as a side note: it is possible that unliganded gating has a very different gating mechanism to the liganded gating, but also that Cys-loop receptors can transition from closed to open states without a ligand and without the loop C capping, but the binding of a ligand and the subsequent loop C capping shift this equilibrium, with the underlying mechanism being the same. I would even argue that similarities between GLIC, which is not activated by small agonists, and other members of the superfamily (as nicely explained by the authors in

rows 114-122) suggest this might be the case.

Indeed!

I appreciate, in particular, the inclusion of additional data in Figs. 5c and 6c. I think the new traces, together with the clearly stated number of traces and cells, clearly illustrate how comparable the kinetics across Cys-loop superfamily is.

Excellent!

I take the authors' point on comparing the structures from different receptors and on AlphaFold and thank them for including the protonation sites on GLIC in the supplementary material – I found this helpful.

We're glad to know!

Overall, we are very grateful to Reviewer #3 for their thorough reading of our submission and their very helpful comments. We feel that the newly added figures and text have made the paper clearer and much more focused.

- What are the noteworthy results?

Pentameric ligand gated ion channels contain agonist binding sites at subunit interfaces within the extracellular domain. Therefore, two subunits contribute to each agonist binding site. The primary binding subunit contains a signature C loop that interacts with agonist. Upon agonist binding this C loop closes in on the opposing subunit, here called capping/closing. Previous studies indicated the capping to function as the mechanical link between agonist binding and channel opening occurring in the transmembrane domain.

In this manuscript, by removing ten amino acids of the signature C loop, the authors still observe functional channels, albeit there is no C loop that could close and thus initiate the cascading domino effect towards channel opening. Therefore, they conclude that loop C capping is not required for the pore to open.

- Will the work be of significance to the field and related fields? How does it compare to the established literature? If the work is not original, please provide relevant references.

The work is original and to the best of my knowledge C-loop deletions have not been published for pentameric channels.

The established literature, also cited by the authors, shows that using structure determination (mostly cryo-EM) that loop C closing behavior is not the same across all pentameric channel subunits.

Citation 10: S. Masiulis, et al., GABAA receptor signaling mechanisms revealed by structural 374 pharmacology. *Nature* 565, 454–459 (2019). GABA in these structures is the agonist and the other ligands are either channel blockers (PTX) or positive allosteric modulators (benzos)

- Figure 5 and other figures in supps: The primary binding site subunit (beta) C-loops close upon the agonist. Other C-loops (alpha, gamma) do not close or do so to a significantly lesser degree. Nonetheless a concerted rearrangement of all subunits and tightening of all subunits and subunit interfaces occurs upon agonist binding in the extracellular domain leading to channel opening.
- PTX – GABA structure shows loop C closure in beta subunits and no other changes as described in the bullet above, the channels are closed in that condition due to PTX functioning as a channel blocker. This structure would show that the loop can close on an agonist in the absence of opening since the channel conformational changes are blocked by the channel blocker and also block ECD concerted constricting conformational changes.
- Benzodiazepines and GABA structures show a loop C (alpha subunit) slight opening, benzos promote channel opening. This structure would show that opening of loop C leads to improved channel opening.

Citation 26: Á. Nemezc, *et al.*, Full mutational mapping of titratable residues helps to identify proton-414 sensors involved in the control of channel gating in the *Gloeobacter violaceus* pentameric ligand-415 gated ion channel. *PLoS Biol.* **15**, e2004470 (2017).

- Several of the amino acids deleted in the GLIC loop-C deletions here are shown to be involved in proton sensing. However, there are many residues all throughout GLIC that contribute to proton sensing and channel opening.

- Does the work support the conclusions and claims, or is additional evidence needed?

I believe the current work does not support the conclusion of the authors.

GLIC-GluCl (same for GLIC-Gly)

- GLIC ECD compared to GLIC ECD without loop C
- Normal channel opening
- Conclusion made in paper: gating does not require the capping of loop C
 - As described above agonist binding to loop C causes loop C closure in small molecule pentameric channels
 - For GLIC proton sites have been described all throughout the ECD and TMD. No single mutation or combination so far has been shown to eliminate proton gating. Therefore, GLIC opening does per se not require an agonist binding to loop C, even though loop C does contain proton sensing involved residues. But they alone are not sufficient.
 - It would not be unreasonable to assume that the concerted conformational changes in the ECD in GLIC without loop C still occur since protons bind at a multitude of sites and whether or not loop C is there just does not make a difference. Since we can reasonably well assume that protons are binding to multiple sites within the ECD, they can be assumed to change the conformation of the ECD which in turn changes the conformation of the TMD. Yes, loop C is not there but all other changes are still likely the same.

Muscle nAChR

- nAChR compared to receptors with one out of five subunits having no C-loop
 - for nAChR the primary site subunit with most contributions to ligand binding is the alpha subunit. This is in contrast to GABAA receptors mentioned above where the beta subunit is the primary agonist binding subunit.
 - note: deleting the primary site alpha subunit loop C was not possible or conducted in this paper because that is the loop that harbors the agonist binding site residues
 - at the same time, I note, that this would be the only two subunits that would have a loop C that would close on agonist binding, as the other intersubunit sites do not bind agonist.
 - note: deleting all three other subunit loop C at the same time led to non-functional channels.
 - This makes us think that overall loop C is important for function in heteropentameric eukaryotic channels.
 - Deleting individual loop C in beta, delta or epsilon subunits
 - Appreciate the effort in demonstrating that for the channels recorded in the single channel patches there is a titratable H at 9' demonstrating that those individual channels do have the C loop deleted subunit; however, this does not mean that in the whole cell currents all or most channels do have the subunits incorporated.
 - We are not expecting the beta, delta or epsilon C loops to close as they do not harbor agonist sites. So why does it matter that their C loop deleted counterparts (if they are quantitatively incorporated in the pentamers) are still functional. It would be the expectation that they would be functional.
- Are there any flaws in the data analysis, interpretation and conclusions? Do these prohibit publication or require revision?
 - I do not think that the paper in its present form warrants publication.

- Is the methodology sound? Does the work meet the expected standards in your field?
 - The experiments and methods are overall sound and meet standards in the field.
- Is there enough detail provided in the methods for the work to be reproduced?
 - yes